# SARS-CoV-2 spike protein variant binding affinity to an angiotensin-converting enzyme 2 fusion glycoproteins

Alicia M. Matthews[1]⊚¤, Thomas G. Biel[1]⊚, Uriel Ortega-Rodriguez[2], Vincent M. Falkowski[1], Xin Bush[3], Talia Faison[1], Hang Xie[2], Cyrus Agarabi[1], V. Ashutosh Rao[1], Tongzhong Ju[1]*

**1** Office of Biotechnology Products, Center for Drug Evaluation and Research, Food and Drug Administration, Silver Spring, Maryland, United States of America, **2** Office of Vaccines Research and Review, Center for Biologics Evaluation and Research, Food and Drug Administration, Silver Spring, Maryland, United States of America, **3** Department of Biomedical and Pharmaceutical Sciences, College of Pharmacy, University of Rhode Island, Kingston, Rhode Island, United States of America

⊚ These authors contributed equally to this work.
¤ Current address: Department of Biology, College of Arts and Science, Case Western Reserve University, Cleveland, Ohio, United States of America
* Tongzhong.Ju@fda.hhs.gov

**Data Availability Statement:** In accordance with the FDA and PLOS ONE policies, the data described in this manuscript is publicly available as presented

## Abstract

Severe acute respiratory syndrome coronavirus-2 (SARS-CoV-2), the causative agent of the Coronavirus disease 2019 (Covid-19) pandemic, continues to evolve and circulate globally. Current prophylactic and therapeutic countermeasures against Covid-19 infection include vaccines, small molecule drugs, and neutralizing monoclonal antibodies. SARS-CoV-2 infection is mainly mediated by the viral spike glycoprotein binding to angiotensin converting enzyme 2 (ACE2) on host cells for viral entry. As emerging mutations in the spike protein evade efficacy of spike-targeted countermeasures, a potential strategy to counter SARS-CoV-2 infection is to competitively block the spike protein from binding to the host ACE2 using a soluble recombinant fusion protein that contains a human ACE2 and an IgG1-Fc domain (ACE2-Fc). Here, we have established Chinese Hamster Ovary (CHO) cell lines that stably express ACE2-Fc proteins in which the ACE2 domain either has or has no catalytic activity. The fusion proteins were produced and purified to partially characterize physicochemical properties and spike protein binding. Our results demonstrate the ACE2-Fc fusion proteins are heavily N-glycosylated, sensitive to thermal stress, and actively bind to five spike protein variants (parental, alpha, beta, delta, and omicron) with different affinity. Our data demonstrates a proof-of-concept production strategy for ACE2-Fc fusion glycoproteins that can bind to different spike protein variants to support the manufacture of potential alternative countermeasures for emerging SARS-CoV-2 variants.

## Introduction

Severe acute respiratory syndrome coronavirus 2 (SARS-CoV-2) is a highly contagious novel beta coronavirus that bears 79% identity to SARS-CoV [1] and has spread across the world

in either the main figures of the manuscript or in the supporting information.

**Funding:** This project was funded by the CDER Advanced & Domestic Manufacturing Initiatives (US FDA), the Office of Pharmaceutical Quality Centers of Excellence (US FDA), and CDER Intramural Research Funding (US FDA). The project was also partially supported by Office of Women's Health/FDA (Xie and Ju), and an appointment to the Research Participation Program at the U.S. Food and Drug Administration administered by the Oak Ridge Institute for Science and Education through an interagency agreement between the U.S. Department of Energy and the U. S. Food and Drug Administration.

**Competing interests:** The authors have declared that no competing interests exist.

leading to the coronavirus disease (COVID-19) pandemic [2, 3]. According to the World Health Organization, at the time of this writing SARS-CoV-2 has caused more than 500 million cases of infections and claimed more than 6 million lives globally since the first confirmed cases in December 2019 [4]. Countermeasures against SARS-CoV-2 infection currently include vaccinations, small molecule drugs, and neutralizing monoclonal antibodies. However, multiple SARS-CoV-2 Variants of Concern (VoC) have emerged with enhanced transmissibility and resistance to existing humoral immunity, and/or increased disease severity, resulting in reduced efficacy or even failure of some countermeasures [5]. The potential emergence of new SARS-CoV-2 variants that can evade countermeasure efficacy remains a global concern.

SARS-CoV-2 infection of human cells is initiated by spike (S) protein binding to angiotensin converting enzyme 2 (ACE2) expressed on host cells, followed by the proteolytic activation of the S protein and subsequent membrane fusion [6, 7]. The S protein is a trimeric transmembrane glycoprotein, and ACE2 is a plasma membrane glycoprotein as a homodimer [6, 7]. The receptor binding domain (RBD) of the S protein has high affinity to human ACE2 and is involved in the overall structural dynamic properties, infectivity, immune evasion, and thus consequent transmissibility of SARS-CoV-2 [8]. The K417N, E484K, and/or N501Y mutations in RBD domain of the alpha, beta, gamma and omicron SARS-CoV-2 VoC are found to impact viral replication and transmissibility, immune evasion, and neutralization by monoclonal antibodies, and infection- or vaccination-derived humoral immunity obtained by natural infection and/or immunization [4, 5, 9–13]. As SARS-CoV-2 continues to evolve, there is high likelihood that a variant capable of escaping current countermeasures will emerge and one potential strategy to counteract the emerging virus is to use a soluble ACE2 as a decoy receptor to compete for S protein binding and sequester shed virus from entering and infecting human cells.

Studies using *in vivo* and *in vitro* models have shown that ACE2 fused with an IgG 1 or 4 Fc region is a potential countermeasure against SARS-CoV-2 VoC [14–20]. Soluble ACE2 has been shown to block SARS-CoV-2 entry and replication by binding to the RBD domain of the S proteins in preclinical models [21]. However, this approach has limitations including the short half-life of soluble ACE2 and the potential adverse events caused by elevating ACE2 enzyme levels [18, 22]. A truncated human ACE2 protein fused to an IgG1 Fc domain (ACE2-Fc) as compared to soluble ACE2 has been reported to increase the half-life in murine serum from 8.5 hours to 29 hours [15]. To reduce potential physiological effects on the renin/ angiotensin system caused by elevated ACE2 activity from ACE2-Fc, protein engineering was performed by mutating two residues (H374N and H378N) within the ACE2 domain to abolish its enzymatic activity while retaining its binding capability to the alpha and beta SARS-CoV-2 VoC [20]. However, the binding affinity of these ACE2 fusion proteins to emerging SARS-CoV-2 variants, such as omicron, remains controversial with studies claiming the binding affinity to ACE2 was enhanced or comparable to the parental SARS-COV-2 variant [23, 24]. Here, we generated a laboratory scale production process to obtain purified ACE2-Fc with and without the inactivation mutations that can be used to test the hypothesis that emerging VoC that evade current countermeasures can be neutralized by ACE2 fusion proteins.

To assess S protein binding affinity to ACE2 fusion proteins, we established Chinese Hamster Ovary (CHO) cell lines that stably express a recombinant fusion protein that is composed of the extracellular domain of human ACE2 with and without the H374N and H378N mutations at the N-terminus and human IgG1 Fc at the C-terminus [ACE2-Fc and ACE2(NN)-Fc] and performed laboratory-scaled protein production. Purified recombinant ACE2-Fc fusion proteins were characterized for stability under thermal stress, N-glycan profiles by mass spectrometry (MS), and binding activity to the S protein variants- parental, alpha, beta, delta, and omicron. We demonstrated that the recombinant ACE2-Fc and ACE2(NN)-Fc possess

binding activity to S proteins. Furthermore, the binding affinity of omicron variant to the fusion proteins is similar to that of the parental S protein. Collectively, these data can be utilized to support the development of ACE2-Fc fusion proteins as potential countermeasures against SARS-CoV-2 for the COVID-19 public health emergency.

## Material and methods

### ACE2-Fc and ACE2(NN)-Fc fusion protein design

To generate the recombinant human ACE2(NN)-Fc protein, the transmembrane domain (amino acids: 741 to 805) within the ACE2 (UniProtKB: Q9BYF1) primary sequence was replaced with a GGGS linker sequence followed by addition of the human IgG1 Fc domain at the C-terminus. The N-terminal endogenous signaling peptide of ACE2 was replaced with a Kappa signaling peptide. The codons related to the histidine amino acids at 374 and 378 were mutated to translate asparagine. The entire primary sequence was reverse translated into DNA and chemically synthesized into a pcDNA3.1/Zeo vector (Genscript). Using molecular cloning techniques, the ACE2(NN)-Fc sequence was cloned into a reporter vector that used an internal ribosomal entry site to drive the expression of green fluorescent protein (GFP) and named ACE2(NN)-Fc-IRES GFP. The wild ACE2 sequence was synthesized and cloned into the ACE2(NN)-Fc-IRES GFP vector to generate an ACE2-Fc-IRES GFP vector.

### CHO cell growth and development

FreeStyle CHO S cells were grown as described by the manufacturer (Cat# R80007, Thermo-Fisher). Cells were electroporated with the ACE2-Fc-IRES GFP or ACE2(NN)-Fc-IRES GFP vectors and subjected to antibiotic selection. One round of Fluorescence Activated Cell Sorting (FACs) was performed to enrich the GFP population. Enriched populations were seeded at 300 cells per well in a 6-well dish using semi solid CHO Growth A medium (Cat # K8810, Molecular Devices) with 7.5 μg of puromycin (Cat# 58-58-2, Sigma). Cell colonies were then imaged, ranked by Fluorescent Total Sum Intensity, picked, and placed in 96-well plates containing FreeStyle CHO Medium (Cat # 12651014, ThermoFisher) using the ClonePix 2 Mammalian Colony Picker as described by the manufacturer (Molecular Devices). The cell population from each clone was counted on days 0 and 4 using a Celigo S cytometer (Nexcelom). On day 4, media from each well was sampled for protein titer via biolayer interferometry using Protein A biosensors and an Octet 96e as described by the manufacture (Sartorius). One colony from each line was selected for expansion, banking, and storage in liquid nitrogen.

### Production of ACE2-Fc and ACE2(NN)-Fc fusion protein cell lines

CHO cell lines expressing ACE2-Fc and ACE2(NN)-Fc were thawed and expanded using 1L shake flasks in an environmentally controlled shake incubator under the following conditions: 5% $CO_2$, 37°C, and 125 rpm. Flasks with FreeStyle CHO Medium (350 mL) were inoculated with $0.3 \times 10^6$ cell/mL for a 7-day unfed or fed batch campaign. For fed batches, the cells were supplemented with bolus additions of 2X CD Efficient Feed C AGT, L-Glutamine, and NaOH on days 3 and 5. L-Glutamine (200 mM/L) was added to maintain a 6 mM/L concentration in the production media. Media pH was adjusted between 7 and 7.4 using 0.5 M NaOH additions. Feed C AGT was added to recover glucose between 4 and 6 mM/L while maintaining the Osmolality < 360 Osm/L. A BioProfile Flex analyzer (Nova Biomedical) was used for all cell culture media nutrient measurements. Batches were sampled on days 3, 5, and 7 to characterize cell viability via trypan blue staining and protein titer via Protein A biosensors (Cat# 18–5010, Sartorius, Germany) and an Octet 96e (Sartorius) as described by the manufacturer.

## Purification of ACE2-Fc and ACE2(NN)-Fc fusion proteins

Harvest cell media was pooled, clarified by centrifugation, and filtered prior to a two-step purification process. HiTrap Protein A HP resin was used to capture the fusion protein from harvest media (Cat# 17040303, Cytiva). The protein A column was washed with 5 column volumes (CV) of 100 mM Sodium Phosphate pH 7.4 buffer prior to elution using 5 CV of a 1 M glycine pH 3 solution. The eluate was concentrated and subjected to a buffer exchange (10 mM sodium chloride and 25 mM Tris pH 8 buffer) using 30 kDa molecular weight cut-off filter (Cat# UFC903024, Millipore). The concentrated eluate was subjected to anion exchange chromatography in bind-and-elute mode using Mono Q 5/50L resin (Cat# 17516601, Cytiva). Using a step gradient, a 350 mM sodium chloride and 25 mM tris pH 7.2 solution was applied to elute the fusion proteins prior to a final buffer exchange into formulation buffer composed of 1% sucrose, 100 mM sodium chloride, 25 mM L-arginine hydrochloride, and 25 mM sodium phosphate buffer at pH 7.4, The final bulk drug substance was concentrated to 1 mg/ml and stored at -80˚C.

## Protein identity, stability, and purity via SDS-page and immunoblotting

SDS-PAGE gels were used to determine the identity, purity, and stability of the proteins (Cat# 4569033, BIO RAD). Protein (1.5 μg) was loaded into each well and following electrophoresis the gel was stained with EZBlue Coomassie Stain (Cat#G1041-500ML, Sigma Aldrich) in accordance with manufacturer protocol. Immunoblotting was performed using 15 microliters of cell media or 0.5 microgram (0.5 μg) of purified protein and the membranes were probed using the following antibodies: Anti-human Fc (Cat# NBP1-40876, Novus Biologicals), ACE2 (Cat# SC-39085, Santa Cruz Biotech), Donkey Anti-Mouse (Cat# 926–68072, Li-cor). Membranes were imaged using Azure 600 imaging system (Azure Biosystems, Dublin, CA).

## ACE2 activity assay on ACE2-Fc and ACE2(NN)-Fc fusion proteins

A commercially available ACE2 activity assay kit was used to measure the activity of the ACE2-Fc and ACE2(NN)-Fc proteins (Cat# MAK377-1KT, Sigma Aldrich). The assay was performed in accordance with the manufacture's protocol.

## N-glycan release and analysis of permethylated N-glycan alditols by MALDI-TOF/TOF-MS

N-glycans from 25 μg of fusion glycoprotein were released using a Filter Aided N-glycan Separation (FANGS) approach with materials from an Abcam Filter Aided Sample Prep (FASP) protein digestion kit (Cat# ab270519, Abcam) as described [25]. Briefly, aliquots containing fusion glycoprotein were reduced with a urea-dithiothreitol (DTT) solution (100mM Tris/HCL pH 8.5, containing 10mM DTT) followed and the mixture was transferred to a 30 kDa molecular cutoff ultrafiltration device and centrifuged at 15,000 x g for 15 minutes. The filter bed was washed repeatedly using urea solution (8 M in 100mM Tris/HCL pH 8.5), and cysteine residues were alkylated with 40 mM iodoacetamide (IAA) for 20 minutes in the dark. The filter was then washed repeatedly with Urea solution, and buffer-exchanged with 50 mM ammonium bicarbonate (pH 7.5). Finally, 50 mM ammonium bicarbonate containing 500U of glycerol-free PNGase F (Cat# P0709S, NEB) was added and the ultra-filtration cell was transferred to a fresh collection tube and incubated at 37˚C for 21 hours. Released N-glycans were recovered by washing the filter LC-MS grade water, followed by centrifugation. Dry N-glycans purified with active charcoal micro spin column (Cat# 74–4800, Harvard Apparatus) as described [26] and the N-glycan eluate was dried by vacuum centrifugation. N-glycans were

reduced with a 10 mg/mL ammonia borane complex in water at 60˚C for 1 hour. The samples were dried under vacuum, followed by a series of methanolic evaporations by adding 10% acetic acid in methanol and drying thoroughly under vacuum until excess reactants were removed. Reduced N-glycans were solubilized in 65 μL anhydrous dimethylsulfoxide, followed by 5 μL of LC-MS water, and 35 μL iodomethane and permethylated by solid-phase sodium hydroxide as described [27]. Permethylated N-glycans were subsequently purified by C18 micro spin columns (Cat# 74–4601, Harvard Apparatus). The final eluate was dried by vacuum centrifugation and resuspended in 50% LC-MS methanol, mixed 1:1 with 2,5-dihydroxybenoic acid (10mg/mL containing 1mM NaCl in 30% ACN, 0.1% TFA) and spotted on a ground steel target. Glycans were analyzed using a Bruker UltrafleXtreme MALDI-TOF/TOF mass spectrometer. Mass measurements were obtained in positive ion reflector mode. Sodiated N-glycan peak intensity was normalized by converting to % abundance based on total glycan identifications.

## Thermal stress induced protein unfolding

Purified ACE2-Fc and ACE2(NN)-Fc (0.2 mg/L) were subjected to thermal ramping between 20˚C and 95˚C at 1˚C increments while monitoring the intrinsic fluorescence using differential scanning fluorimetry (Uncle, Unchained Labs) in accordance with the manufacturer's protocol.

## Biolayer interferometry

Anti-human IgG Fc capture biosensors (Cat# 18–5060, Sartorius) were equilibrated for 10 minutes in fresh FreeStyle CHO Medium prior to performing BLI. Using an Octet96e, ACE2-Fc and ACE2(NN)-Fc were immobilized on to the biosensors by placing the tip of the biosensor in a solution that contained 500 ng/mL ACE2-Fc or ACE2(NN)-Fc in FreeStyle CHO Medium for 5 minutes at 25˚C. The loaded tips were incubated with the S protein variants (Cat # 40594-V08H,0591-V08H10, 40594-V08H12, 40591-V08H19, Sinobiological and Cat# S1n-C52Ha Acrobiosystems) at different concentrations (0, 0.06 0.1375, 0.375, 0.75, 1.5, 3, and 6 mg/mL) for 400 seconds to determine the association rate and protein binding affinity. Next, the biosensors placed in fresh FreeStyle CHO Medium for 400 seconds to determine the dissociation rate. Fit curves with an R squared of greater than 0.95 were used to determine the KD (M), association rate, and dissociation rate using Data Acquisition Software (Version 12.0.2.11).

## Statistics and reagents

All other reagents were purchased from Sigma Aldrich. Statistical analyses were performed using GraphPad Prism 6 from at least two independently performed experiments. When 1 factor was analyzed, a one-way ANOVA was performed with a Sidak's multiple comparison test to identify statistical differences. When >1 factor was analyzed, a two-way ANOVA was performed with a Tukey's multiple comparison post-hoc test to identify significant differences. A $p$ value of $< 0.05$ indicated statistical differences. The degree of freedom, p values, and test methods are provided in S1 Table in S2 File.

## Results

### Upstream processing of CHO cells stably expressing ACE2 fusion proteins

To confirm the expression of ACE2-Fc and ACE2(NN)-Fc proteins from the synthetically derived plasmids, cell free media from CHO cell lines transfected with the ACE2-Fc and ACE2 (NN)-Fc plasmids were subjected to anti-IgG1-Fc and anti-ACE2 immunoblotting. Under

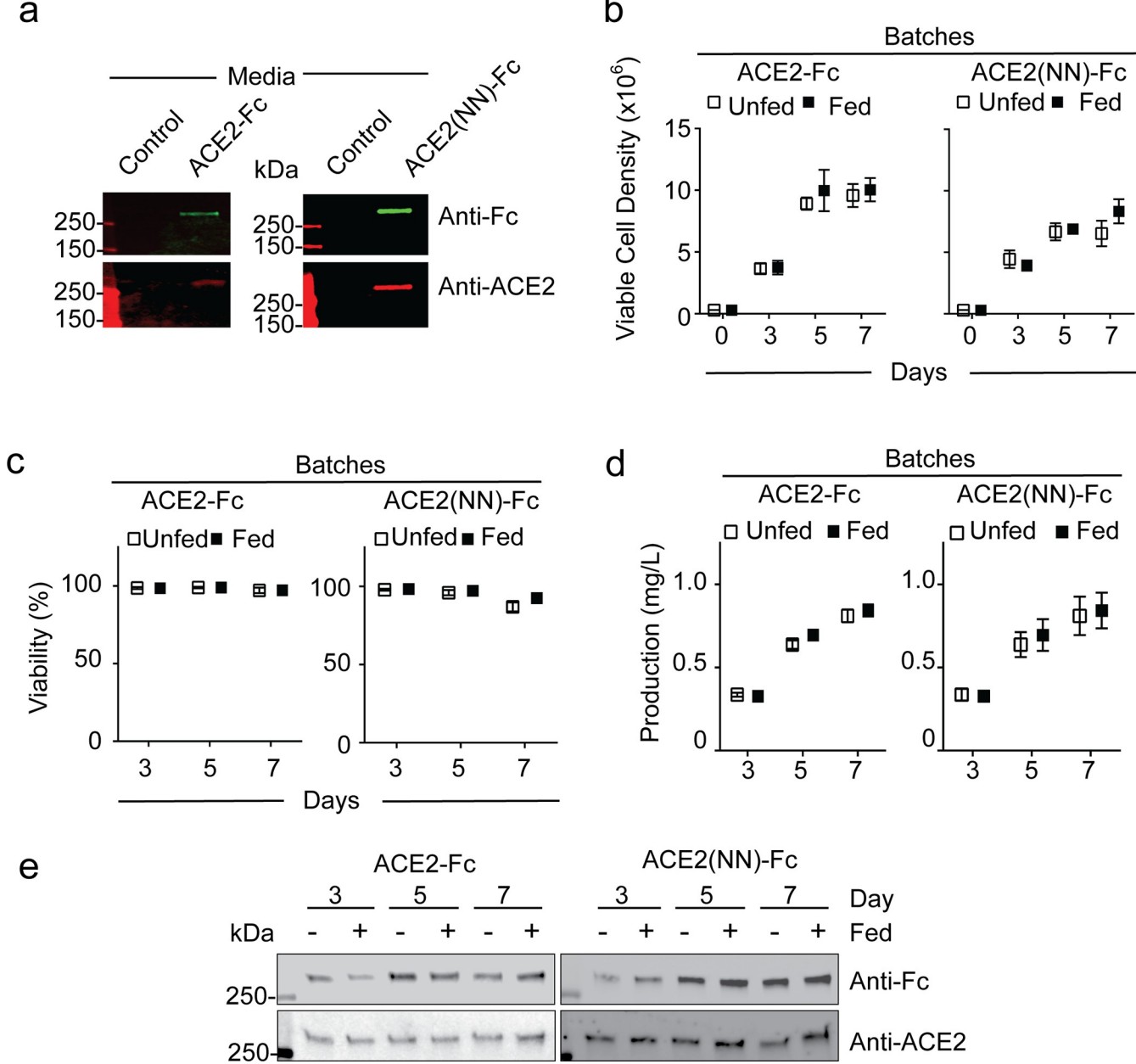

**Fig 1. Upstream production of ACE2-Fc and ACE2(NN)-Fc using stable CHO cell lines.** (**a**) Representative IgG-Fc and ACE2 immunoblots using harvest media (15 µL) compared to blank media. CHO cell viable density (**b**) and viability (**c**) over a 7-day campaign using fed and unfed cultivation (n = 3). (**d**) Protein titer of the ACE2-Fc and ACE2(NN)-Fc during the campaigns under fed and unfed conditions (n = 3). (**e**) Representative IgG-Fc and ACE2 immunoblots using cell free harvest media (15 µL) collected on day 3, 5, and 7. Dots represent mean ± standard deviation.

non-reducing conditions, a protein band greater than 250 kDa was observed in the media of transfected CHO cell lines (Fig 1A). Since the ACE2 fusion proteins have a theoretical molecular weight of approximately 225 kDa as a dimer and 112.5 kDa as a monomer [28], the immunoblotting data confirms the expression of ACE2-Fc and ACE2(NN)-Fc fusion proteins. Following confirmation of the ACE2 fusion protein expression, the cells were subjected to antibiotic selection and monoclonal derivatization to generate CHO cell lines that stably express ACE2 fusion proteins for production.

Continuous and fed-batch cultivation are common upstream processes that increase thera-peutic protein productivity from CHO cells [29]. Fed-batch cultivation involves the addition of feed medium and supplements to the basal medium during production campaigns to facili-tate extending the productive growth phase of CHO cells and improve therapeutic protein pro-ductivity [29]. To assess the capability of the CHO cell lines expressing ACE2-Fc and ACE2 (NN)-Fc to respond to environmental stimuli, the media was supplemented with bolus addi-tions of a glucose with amino acids, L-glutamine, and sodium hydroxide to adjust the pH dur-ing the production campaigns. The bolus additions had a statistically insignificant impact on viable cell density, cell viability, and protein production in these cell substrates (Fig 1B–1D). Over the 7-day campaigns with and without the fed strategy on days 3 and 5, the monoclonal-derived CHO cell lines expressing ACE2-FC and ACE2(NN)-Fc had comparable viable cell density (Fig 1B), total cell viability (Fig 1C), and ACE2 fusion protein production (Fig 1D). To assess batch to batch variability, the viable cell density, cell viability, and protein production were monitored and revealed to be statistically comparable between batches (S1 Fig in S2 File). To confirm the production and secretion of ACE2-Fc and ACE2(NN)-Fc from the monoclo-nal-derived CHO cell lines, cell free harvest media from day 3, 5, and 7 was collected for anti-IgG1 Fc and anti-ACE2 immunoblotting. A protein > 250 kDa was identified in the media that contained both the ACE2 and IgG-Fc domains (Fig 1E). Collectively, these data confirm that the CHO cell lines produced and secreted the ACE2 fusion proteins.

## Purified ACE2 fusion proteins and ACE2 activity

To purify the ACE2-Fc proteins from the cell free harvest media, a two-step chromatography method was performed using Protein A resin to capture the fusion proteins followed by anion exchange chromatography for further removal of impurities. Coomassie-stained SDS-PAGE gels (Fig 2A) showed a main protein band with an apparent molecular weight (MW) > 250 kDa under non-reducing conditions and approximately 150 kDa under reducing conditions, demonstrating that the ACE2-Fc and ACE2(NN)-Fc Proteins are homodimers. To confirm the identity of the purified ACE2 fusion proteins, anti-ACE2 and anti-human IgG Fc immuno-blotting was performed and revealed the purified proteins contained both the ACE2 and IgG1-Fc domains (Fig 2B). The catalytic activity of purified ACE2 fusion proteins was tested and revealed ACE2-Fc had a catalytic capability of 1468 ± 200 mU/mg, while ACE2(NN)-Fc had an activity of -0.1 ± 4.588 mU/mg, which was lower than the assay's limit of detection (Fig 2C). These data qualified the upstream and two-step downstream laboratory-scaled produc-tion process for ACE2-Fc and ACE2(NN)-Fc.

## Glycosylation of ACE2-Fc fusion proteins

Human ACE2 is a homodimer, and each monomer has 7 potential N-glycosylation sites which are occupied predominantly by complex-type N-glycans bearing terminal sialic acid moieties [30]. Each subunit of the ACE2-Fc fusion proteins is predicted to contain 8 potential N-glyco-sylation sites by counting an additional N-glycosylation at the IgG1 Fc [18]. To probe for the presence of N-glycans on ACE2-Fc and ACE2(NN)-Fc, we treated both fusion glycoproteins with peptide-N-glycosidase F (PNGase F), which removes all N-glycan species except for those that carry α1,3 core fucosylation found in invertebrates. Under reducing conditions on SDS-PAGE, untreated ACE2-Fc and ACE2(NN)-Fc migrated at approximately 150 kDa and shifted to approximately 120 kDa after PNGase F treatment (Fig 3A and 3B). These data dem-onstrate that ACE2 fusion proteins are highly glycosylated proteins and the difference between the apparent and theoretical molecular weights are in part due to N-glycosylation of the fusion proteins.

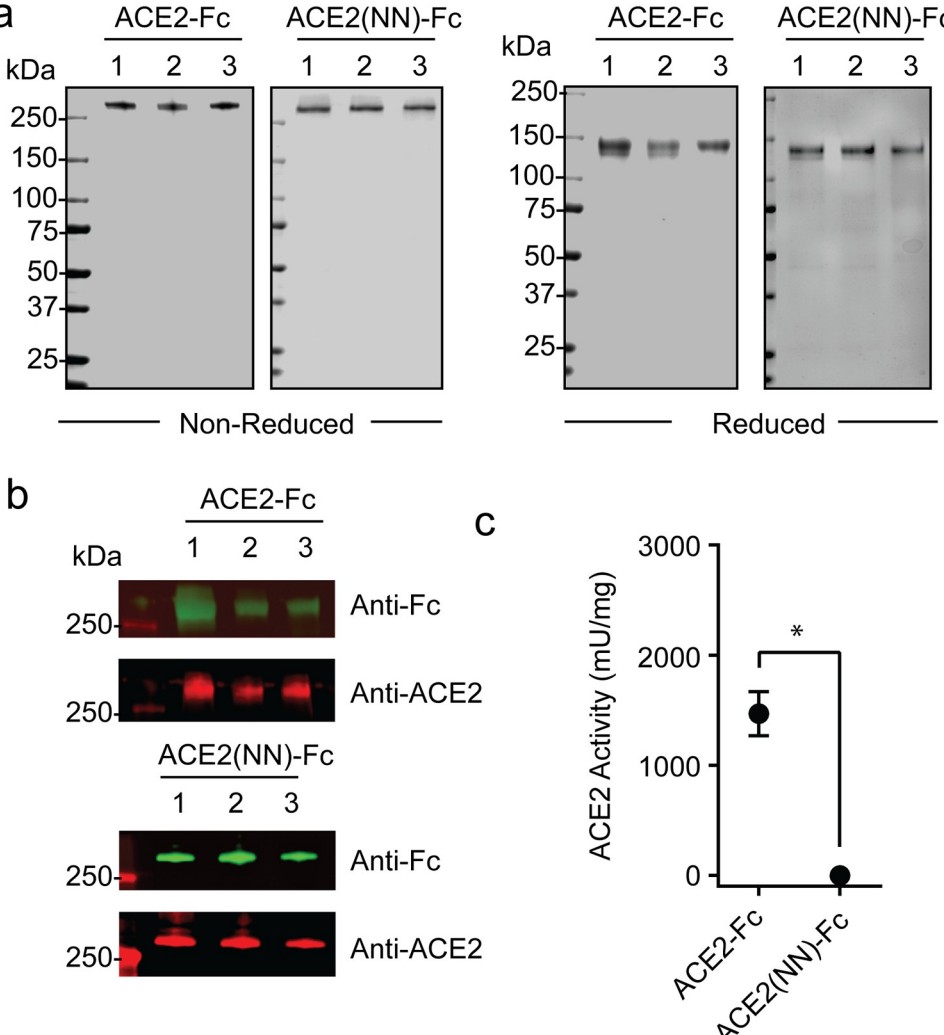

**Fig 2. Purity, identity, and activity characterization of purified ACE2-Fc and ACE2(NN)-Fc. (a)** Coomassie-stained SDS-PAGE gels under non reducing and reducing conditions of 1.5 μg of purified ACE2-Fc and ACE2(NN)-Fc proteins. **(b)** Anti-IgG-Fc and anti-ACE2 immunoblots of purified ACE2-fusion proteins using 0.5 ug of protein under non-reducing conditions. **(c)** ACE2 enzymatic activity of purified ACE2-Fc and ACE2(NN)-Fc proteins (n = 4). Two batch of ACE2-Fc and ACE2(NN)-Fc was performed in quadruplicate. Dots represent mean ± standard deviation and * indicates *p* <0.05.

To characterize the relative abundance of unique glycan structures on ACE2-Fc and ACE2(NN)-Fc, N-glycans were released by PNGase F, reduced, and permethylated for analysis by mass spectrometry (MS). In total, 24 unique peaks consistent with monoisotopic *m/z* values of sodiated N-glycan species were identified in both ACE2-Fc and ACE2(NN)-Fc (Fig 3C). Of the 24 N-glycan species with different compositions observed, 19 (79.17%) contained terminal sialic acid moieties but no hybrid or high mannose types were observed. Notably, several asia-lylated biantennary species were identified; FA2 (1851 *m/z*) being the predominant glycan which most likely occupied the Fc domain of ACE2-Fc and ACE2-(NN)-Fc as seen in CHO-produced IgG1-Fc [31]. The second dominant N-glycan, FA2G2S2 (2982 *m/z*), carried 2 Neu5Ac residues and most likely occupied the N-glycosylation sites within the ACE2 domain of the fusion proteins [31]. We also observed core fucosylation in 87.5% (21/24) of the N-

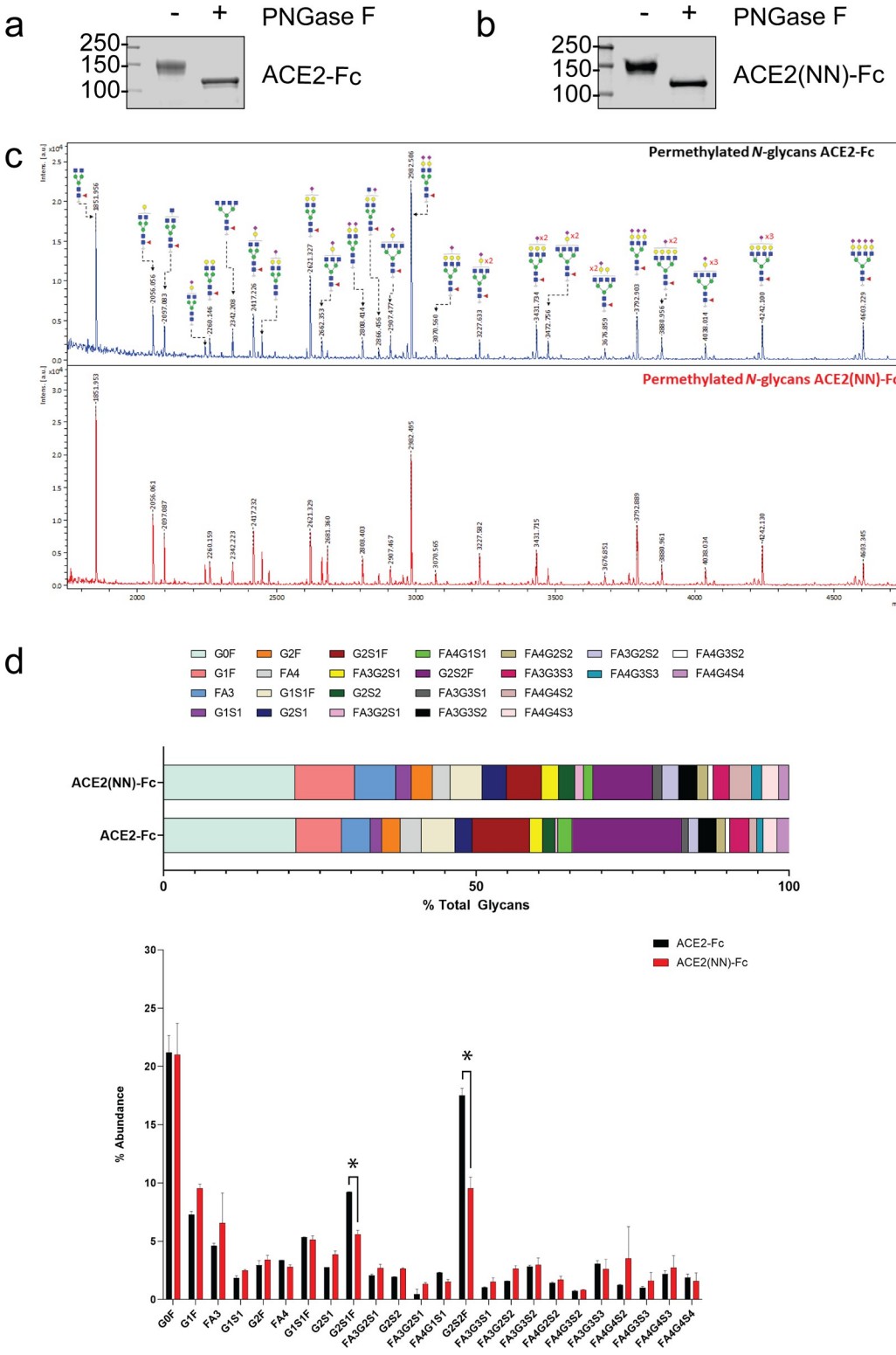

**Fig 3. ACE2-Fc and ACE2(NN)-Fc fusion proteins are heavily glycosylated with complex-type N-glycans.** Reduced SDS-page of ACE2-Fc (**a**) and ACE2(NN)-Fc (**b**) before and after treatment with PNGase F. (**c**) MALDI-TOF-MS spectra of permethylated N-glycans identified in ACE2-Fc (top) and ACE2(NN)-Fc(bottom). (**d**) Relative abundance of N-glycan peaks identified in ACE2-Fc and ACE2(NN)-Fc (Top), and comparison in percentage abundance of N-glycans identified in 3 batches of ACE2-Fc and 3 batches of ACE2(NN)-Fc (bottom).

glycan species, and in 92.21% relative abundance which is consistent with the abundant core fucosylated glycoproteins expressed in CHO cell lines. In addition, tri-antennary and tetra-antennary N-glycans with variable sialylation were also observed which is consistent with endogenous N-glycan profiles of CHO cells [32]. These highly branched complex N-glycans were also likely to be presented within the ACE2 domain. Non-human glycoepitopes, such as α-Gal or Neu5Gc, were not detected in the ACE2-Fc fusion proteins. The comprehensive N-glycan profile of 3 batches of ACE2-Fc and ACE2(NN)-Fc was obtained to monitor batch-to-batch glycosylation consistency (S2 Table in S2 File). Side-by-side comparison of relative abundance of each glycan showed a statistically significant reduction of GS2S1F and GS2SF in ACE2(NN)-Fc as compared to ACE2-Fc (Fig 3D) while comparable for other 22 N-glycan species between ACE2-Fc and ACE2(NN)-Fc. These data confirm that ACE2-Fc and ACE2(NN)-Fc are heavily N-glycosylated and bear a broad variety of complex-type N-glycans.

## Stability of ACE2 fusion proteins subjected to thermal stress

Therapeutic protein instability can directly impact the manufacturing strategy [33] and patient safety [34]. To assess the stability of the ACE2 fusion proteins, purified proteins in the storage buffer formulation were subjected to different stress conditions to monitor for protein instability as compared to the fusion protein stored at -80˚C. As shown in Fig 4A, ACE2-Fc and ACE2(NN)-Fc at 4˚C for 1 and 7 days or 25˚C for 1 day did not demonstrate gross changes in protein fragmentation or aggregation, but the protein band was completely absent after 7 days at 25˚C. These data indicate that ACE2-Fc and ACE2(NN)-Fc proteins are sensitive to thermal-induced stress.

Temperature-induced protein unfolding using differential scanning fluorimetry was performed to assess changes in higher order structures. Over 1-degree Celsius increments, the intrinsic fluorescence intensity of the ACE2-FC and ACE2(NN)-Fc was observed to decrease between 295 and 395 nm (Fig 4B and 4C). Using the change in barycentric mean of the fluorescent intensity at each temperature, the protein melting temperature (Tm) curves were calculated for the ACE2 fusion proteins (Fig 4D and 4E). The Tm of ACE2-Fc and ACE2(NN)-Fc was 57.3 ± 11.6˚C and 48.1 ± 0.25˚C, respectively. T onset is the temperature that a protein begins to unfold, which for ACE2-FC and ACE2(NN)-Fc was 42.7 ± 2.9˚C and 40.7 ± 4.3˚C, respectively. These data indicate the higher order structure of the ACE2 fusion proteins in storage buffer formulation are sensitive to prolonged exposure to 25˚C and temperatures above approximately 40˚C.

## Binding affinity of ACE2-Fc and ACE2(NN)-Fc to spike protein variants

Biolayer interferometry (BLI) was performed to determine the binding affinity, disassociation rate, and association rate of ACE2-Fc and ACE2(NN)-Fc to five different S protein variants: parental, alpha, beta, delta, and omicron. All five S protein variants exhibited binding activity to immobilized ACE2-Fc and ACE2(NN)-Fc (Fig 5A and 5B). Alpha, beta, and delta S proteins had an enhanced binding affinity to both ACE2 fusion proteins as compared to the parental and omicron S protein variants (Fig 5C and Table 1). The alpha S protein variant had a significantly slower dissociation rate from the ACE2 fusion proteins (Fig 5D and Table 1), while the association rate between the S protein variants and the ACE2 fusion proteins remained comparable to the parental variant. Collectively, these data demonstrated that ACE2-Fc and ACE2(NN)-Fc fusion proteins are active in binding S proteins, more importantly, with greater affinities for the alpha, beta, and delta S protein variants as compared to the parental and omicron variant.

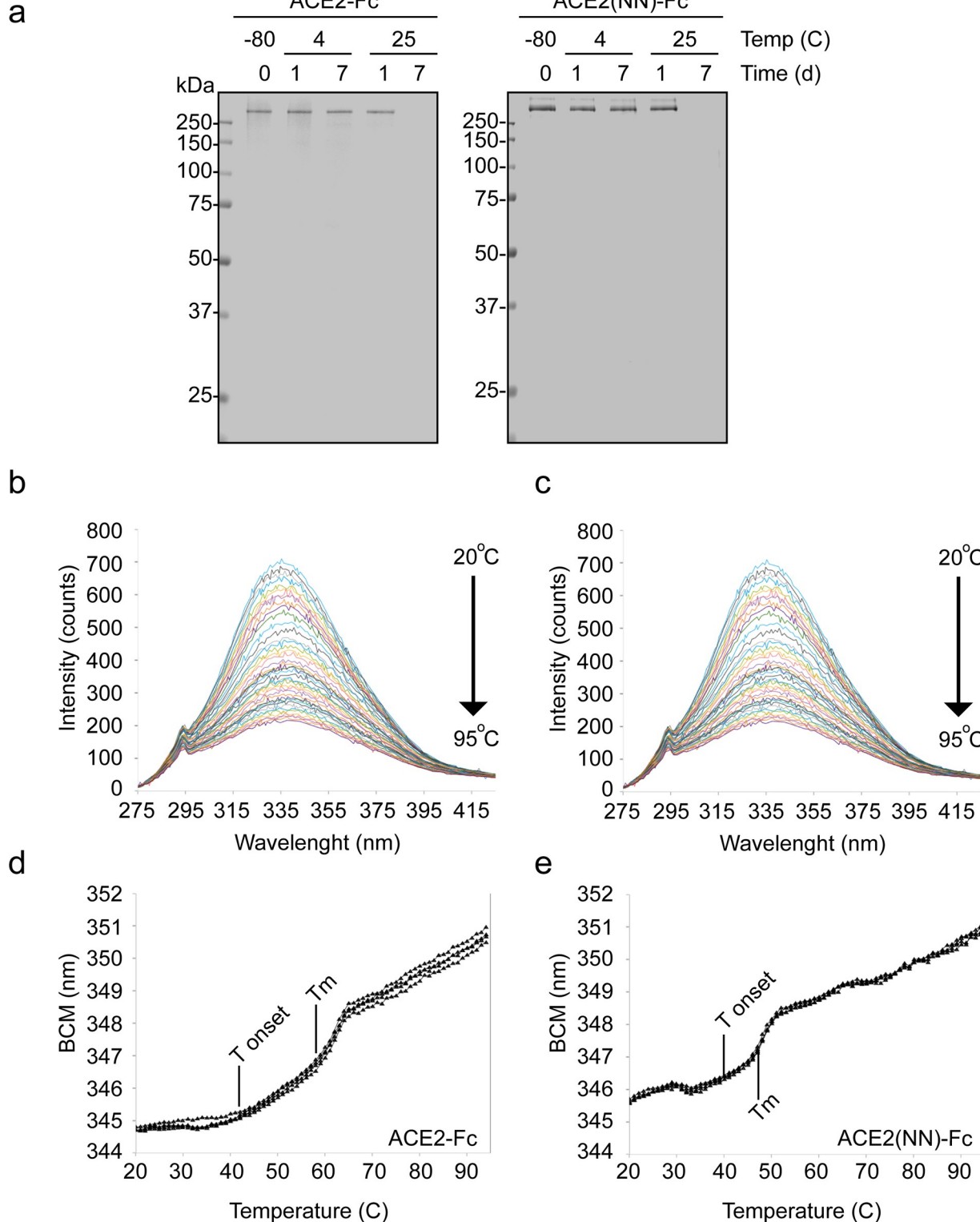

**Fig 4. ACE2-Fc and ACE2(NN)-Fc thermal stability assessment.** (**a**) Non-reduced Coomassie-stained SDS-page gels of the ACE2-Fc fusion proteins (1.5 μg) in the storage condition (-80˚C) and at 4˚C and 25˚C for 1 and 7 days. Representative spectra of intrinsic fluorescent intensity curve changes of ACE-Fc (**b**) and ACE2(NN)-Fc (**c**) over a 20˚C to 95˚C temperature range at 1˚C increments. The barycentric mean (BCM) of the intrinsic fluorescent curves was plotted against the temperature to determine onset of protein unfolding (T onset) and melt temperature (Tm) of ACE2-Fc (**d**) and ACE2(NN)-Fc (**e**). One batch of each ACE2 fusion protein was analyzed in quadruplicate (n = 4).

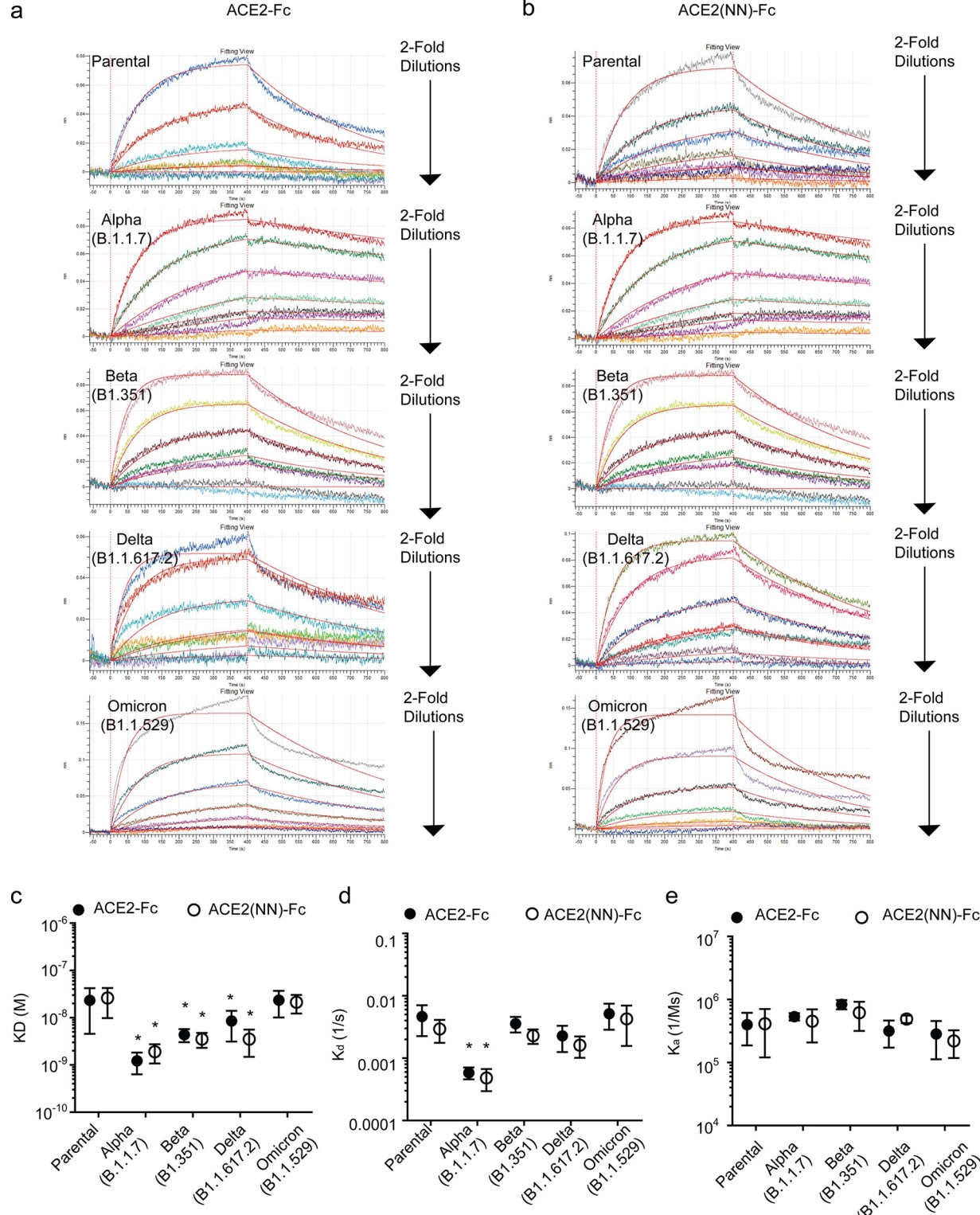

**Fig 5. ACE2-Fc and ACE2(NN)-Fc binding affinity to S protein variants using biolayer interferometry.** Representative electropherograms of ACE2-Fc (**a**) and ACE2(NN)-Fc (**b**) binding to S protein variants: parental, alpha, beta, delta, and omicron. The binding affinity (**c**), disassociation rate (**d**), and association rate (**e**) of ACE2-Fc to S protein variants: parental, beta, alpha, delta, and omicron. Three batches of ACE2 fusion proteins were analyzed in triplicate (Two Way ANOVA, Factors: Variant and ACE2 fusion protein, n = 9). Dots represent the mean and bars represent the 95% Cl. * indicate p < 0.05 within the ACE2-Fc or ACE2(NN)-Fc groups.

**Table 1. Average ACE2-Fc and ACE2(NN)-Fc binding affinity, association rates, and dissociation rates with different spike protein variants.**

| | ACE2-Fc | | | | | |
|---|---|---|---|---|---|---|
| Spike protein variant | KD (nM) | S.D. | Ka (1/Ms) | S.D. | Kd (1/S) | S.D. |
| Parental | 23.35 | 21.45 | 4.02E+05 | 1.59E+05 | 4.66E-03 | 3.13E-03 |
| Alpha (B.1.1.7) | 1.23 | 0.78 | 5.31E+05 | 7.55E+04 | 5.83E-04 | 1.63E-04 |
| Beta (B1.351) | 4.38 | 1.74 | 8.40E+05 | 1.42E+05 | 3.59E-03 | 1.31E-03 |
| Delta (B1.1.617.2) | 8.54 | 7.06 | 3.18E+05 | 1.45E+05 | 2.28E-03 | 1.34E-03 |
| Omicron (B1.1.529) | 23.59 | 17.53 | 2.84E+05 | 1.71E+05 | 5.17E-03 | 2.99E-03 |
| | ACE2(NN)-Fc | | | | | |
| Spike protein variant | KD (nM) | S.D. | Ka (1/Ms) | S.D. | Kd (1/S) | S.D. |
| Parental | 26.17 | 21.33 | 4.13E+05 | 2.92E+05 | 2.94E-03 | 1.53E-03 |
| Alpha (B.1.1.7) | 1.91 | 1.08 | 4.53E+05 | 2.45E+05 | 4.82E-04 | 2.43E-04 |
| Beta (B1.351) | 3.53 | 1.61 | 6.19E+05 | 3.02E+05 | 2.29E-03 | 7.79E-04 |
| Delta (B1.1.617.2) | 3.55 | 2.46 | 5.03E+05 | 5.84E+04 | 1.62E-03 | 7.93E-04 |
| Omicron (B1.1.529) | 21.30 | 11.75 | 3.77E+05 | 3.02E+05 | 4.28E-03 | 3.51E-03 |

S.D. = Standard Deviation.

## Discussion

This report describes the generation of CHO cell lines that stably express ACE2-Fc and ACE2 (NN)-Fc for SARS-CoV-2 research and the qualification of an upstream and downstream process to obtain purified ACE2 fusion proteins. Using the purified ACE2-Fc and ACE2(NN)-Fc fusion proteins, characterization studies were performed to assess N-glycosylation, thermal stability, and S protein variant binding affinity. Our results are the first to report the N-glycosylation profile of ACE2-Fc and ACE(NN)-Fc, the instability of ACE2 fusion proteins under thermal stress in a sucrose-based storage buffer, and a binding affinity comparative analysis within five different S protein variants: parental, alpha, beta, delta, and omicron. This study can be leveraged to support new ACE2 fusion protein designs, purification strategies, and formulation studies to facilitate the manufacture of ACE2-Fc and ACE2(NN)-Fc for potential studies related to the COVID-19 public health emergency, and potentially other ACE2-dependent coronavirus diseases.

ACE2-Fc has been reported as a potent prophylactic and treatment for COVID-19 based on *in vivo* and *in vitro* studies [14, 15, 17, 18, 20]. However, several physicochemical properties of the ACE2 fusion proteins remain elusive and require characterization. For example, N-glycosylation is a critical quality attribute on many protein drugs, such as monoclonal antibodies, due to their potential impact on drug safety and efficacy [35]. N-glycans on endogenous ACE2 at residues N90 and N322 are claimed to have opposing effects on spike protein binding [36]. It is reported that glycosylation at N90 on ACE2 can impair S protein docking, while glycans at N322 of ACE2 can strengthen the association with the RBD of the S protein [36]. This suggests that N-glycosylation is likely to be a critical quality attribute for potential ACE2 fusion protein therapies. Here, the ACE2-Fc and ACE2(NN)-Fc proteins produced from CHO cells were identified to be heavily N-glycosylated. The differences in glycans between ACE2-Fc and ACE2(NN)-Fc could be due to mutations introduced in ACE2(NN)-Fc, or the cell clone phenomenon, which need further studies. To further determine the impact of these glycans on the potency and efficacy of the fusion proteins, more studies are needed for characterizing the N-glycans for each glycosite, followed by N-glycan engineering to evaluate S protein binding.

Long-term storage and accelerated stress stability studies are required to initiate clinical studies and support storage of therapeutic proteins [37]. The fusion of the IgG1-Fc region to the ACE2 domain has been shown to increase serum stability as compared to ACE2 [22], but

the stability of the fusion protein under different storage and accelerated stress conditions remains limited. Using a 20 mM histidine only formulation, the Tm of ACE2(NN)-Fc was determined to be pH dependent with a range from 42.3˚C at pH 3.5 to 51.6˚C at pH 6.5 [20]. The non-histidine-based storage buffer (1% sucrose, 100 mM sodium chloride, 25 mM L-arginine hydrochloride, and 25 mM Sodium Phosphate buffer at pH 6.3) used in this report had a comparable melting temperature for ACE2(NN)-Fc (48.1 ± 0.25˚C) and a potentially comparable to slightly higher melting temperature for ACE2-Fc (57.3 ± 11.6˚C), suggesting the wild-type ACE2 is more stable than mutated ACE2 at H374N and H378N. These data can be leverage for additional studies focusing on the content of the formulation to improve the stability of the ACE2 fusion proteins.

Several studies have investigated the binding affinity between RBD domain of VoC S protein and ACE2 fusion proteins [14, 15, 20]. Our ACE2-Fc fusion proteins with and without ACE2 enzymatic activity had a similar binding affinity to the full-length S protein variants tested, parental, alpha, beta, delta, and omicron. Furthermore, alpha, beta and delta S proteins had an enhanced binding affinity to both ACE2 fusion proteins as compared to the parental and omicron S protein variants. The relative binding affinity of the ACE2 fusion proteins to the emerging SARS-CoV-2 omicron variant remain controversial: some studies claimed to be enhanced [14, 23] while other showed to be comparable [24] as compared to the parental SARS-CoV-2 variant. Our results showed that the affinity of full-length S protein omicron variant to the ACE2 fusion proteins was comparable to the parental strain. The claim from previous reports that the omicron variant had enhanced binding affinity to the ACE2-Fc proteins lacked support from repeated or replicated assays and from a validated method. Our conclusion was drawn from three independently performed experiments for each of the three different fusion protein batches using a validated assay with inter-assay precision and reproducibility.

Binding affinity of the ACE2 fusion proteins is an important physicochemical property, but a cell-based potency assay is warranted to assess biological functions or potency. Studies have demonstrated ACE2-Fc can neutralize the SARS-CoV-2 virus using *in vitro* and *in vivo* models [14, 15]. However, we were unable to assess the neutralization of the SARS-CoV-2 virus due to insufficient quantities of purified recombinant fusion protein caused by the limitations in our laboratory-scaled production system, which were primarily due to the low productivity of the cell substrates, the inefficiency of the downstream purification, and the instability of the protein in our current storage formulation. Based on the limitations in this production process, it was deemed appropriate to only perform affinity binding studies, while new cell substrates that produce a greater quantity of ACE2 fusion proteins are developed for addressing the downstream and protein stability limitations.

## Conclusions

As the SARS-CoV-2 continues to circulate and mutate, there is a global concern that an emerging variant will evade countermeasure efficacy. A therapeutic fusion protein designed to leverage the mechanism of action for preventing SARS-COV-2 infections of host cells has not yet been clinically studied but may have the potential to be an alternative approach to counter SARS-CoV-2, in addition to vaccinations, small molecule drugs and neutralizing mAbs. This study provides data supporting ACE2 fusion proteins as potential therapeutics by analyzing their glycosylation, stability, expression, and binding affinity to SARS-CoV-2.

## Supporting information

**S1 File. Immunoblots, gels, and quantitative results as supportive data.**
(PDF)

**S2 File. Supplementary tables and figure.**
(PDF)

## Acknowledgments

We would like to thank Elliot Rosen and Dr. Su-ryun Kim for critically reviewing the manuscript.

## Author Contributions

**Conceptualization:** Thomas G. Biel, Tongzhong Ju.

**Data curation:** Alicia M. Matthews, Thomas G. Biel, Uriel Ortega-Rodriguez, Tongzhong Ju.

**Formal analysis:** Alicia M. Matthews, Thomas G. Biel, Uriel Ortega-Rodriguez, Hang Xie, Cyrus Agarabi, V. Ashutosh Rao, Tongzhong Ju.

**Funding acquisition:** Thomas G. Biel, Hang Xie, Cyrus Agarabi, V. Ashutosh Rao, Tongzhong Ju.

**Investigation:** Alicia M. Matthews, Thomas G. Biel, Uriel Ortega-Rodriguez, Vincent M. Falkowski, Xin Bush, Talia Faison, Hang Xie, Cyrus Agarabi, V. Ashutosh Rao, Tongzhong Ju.

**Methodology:** Alicia M. Matthews, Thomas G. Biel, Uriel Ortega-Rodriguez, Xin Bush, Talia Faison, Tongzhong Ju.

**Project administration:** Thomas G. Biel, Tongzhong Ju.

**Resources:** Thomas G. Biel, Tongzhong Ju.

**Supervision:** Thomas G. Biel, Tongzhong Ju.

**Validation:** Alicia M. Matthews, Thomas G. Biel, Uriel Ortega-Rodriguez, Cyrus Agarabi, Tongzhong Ju.

**Visualization:** Thomas G. Biel, Tongzhong Ju.

**Writing – original draft:** Alicia M. Matthews, Thomas G. Biel, Uriel Ortega-Rodriguez, Vincent M. Falkowski, Xin Bush, Talia Faison, Hang Xie, Cyrus Agarabi, V. Ashutosh Rao, Tongzhong Ju.

**Writing – review & editing:** Alicia M. Matthews, Thomas G. Biel, Uriel Ortega-Rodriguez, Vincent M. Falkowski, Xin Bush, Talia Faison, Hang Xie, Cyrus Agarabi, V. Ashutosh Rao, Tongzhong Ju.

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
