## [Decision Letter · Decision Letter 0]

19 Oct 2022

PONE-D-22-26644SARS-CoV-2 Spike variants binding affinity to an Angiotensin-converting enzyme 2 fusion glycoproteinPLOS ONE

Dear Dr. Ju,

Thank you for submitting your manuscript to PLOS ONE. After careful consideration, we feel that it has merit but does not fully meet PLOS ONE’s publication criteria as it currently stands. Therefore, we invite you to submit a revised version of the manuscript that addresses the points raised during the review process.

We look forward to receiving your revised manuscript.

Kind regards,

Nagarajan Raju

Academic Editor

PLOS ONE

“This project was funded by the CDER Advanced & Domestic Manufacturing Initiatives (US FDA), the

Office of Pharmaceutical Quality Centers of Excellence (US FDA), and CDER Intramural Research

Funding (US FDA). The project was also partially supported by Office of Women’s Health/FDA (Xie and

Ju), and an appointment to the Research Participation Program at the U.S. Food and Drug Administration

administered by the Oak Ridge Institute for Science and Education through an interagency agreement

between the U.S. Department of Energy and the U.S. Food and Drug Administration.”

“This project was funded by the CDER Advanced & Domestic Manufacturing Initiatives (US FDA), the Office of Pharmaceutical Quality Centers of Excellence (US FDA), and CDER Intramural Research Funding (US FDA). The project was also partially supported by Office of Women’s Health/FDA (Xie and Ju), and an appointment to the Research Participation Program at the U.S. Food and Drug Administration administered by the Oak Ridge Institute for Science and Education through an interagency agreement between the U.S. Department of Energy and the U.S. Food and Drug Administration.”

“None.”

Reviewers' comments:

Reviewer's Responses to Questions

**Comments to the Author**

1. Is the manuscript technically sound, and do the data support the conclusions?

Reviewer #1: Yes

Reviewer #2: Yes

2. Has the statistical analysis been performed appropriately and rigorously? 

Reviewer #1: N/A

Reviewer #2: N/A

3. Have the authors made all data underlying the findings in their manuscript fully available?

Reviewer #1: Yes

Reviewer #2: Yes

4. Is the manuscript presented in an intelligible fashion and written in standard English?

Reviewer #1: Yes

Reviewer #2: Yes

5. Review Comments to the Author

Reviewer #1: The article is well written and the results presented appear valid. There is a point I find to be assessed under discussion section.

The activity assay for ACE2 is limited to the specific enzymatic activity. However, ACE2 is involved in a number of protein-protein interactions into the cell, whose function is at least in part unclear. The activity assay can not evaluate the ability of the recombinant fusion protein to mimic ACE2 for these interactions and activities in the cell, so this point should be considered as a risk of interference with cellular processes, with unknown effects.

Reviewer #2: The authors present a well written, concise manuscript describing the use of ACE2 fused to Fc regions as an initial step in developing therapeutics that would act as a "sink" for SARS-2 and potentially SARS, though the authors make no claims about SARS. The authors demonstrate that a couple of key mutants knock out the enzymatic function which improves the usability of the constructs as virus sinks. There are a few minor typos that can be easily corrected with another round of proofreading.

Specific comments:

Does the glycan profile vary based on cell type? If the constructs were made in human/nonhuman primate cell lines, would the binding affinities be the same?

How does the ACE2-Fc-virus complex get metabolized to clear virus from the body to prevent infection?

What is the significance of the alpha, beta, and delta having higher affinity than the parental or omicron.

6. PLOS authors have the option to publish the peer review history of their article (what does this mean?). If published, this will include your full peer review and any attached files.

Reviewer #1: No

Reviewer #2: No

---

## [Author Response · Author response to Decision Letter 0]

10 Nov 2022

PONE-D-22-26644

The below Author’s Responses address the PLOS ONE editorial staff’s comments.

Author’s Response: We have revised the manuscript based on the style and information provided in the PLOS ONE style templates.

2) Thank you for stating the following in the Acknowledgments Section of your manuscript:

“This project was funded by the CDER Advanced & Domestic Manufacturing Initiatives (US FDA), the Office of Pharmaceutical Quality Centers of Excellence (US FDA), and CDER Intramural Research Funding (US FDA). The project was also partially supported by Office of Women’s Health/FDA (Xie and Ju), and an appointment to the Research Participation Program at the U.S. Food and Drug Administration administered by the Oak Ridge Institute for Science and Education through an interagency agreement between the U.S. Department of Energy and the U.S. Food and Drug Administration.”

“This project was funded by the CDER Advanced & Domestic Manufacturing Initiatives (US FDA), the Office of Pharmaceutical Quality Centers of Excellence (US FDA), and CDER Intramural Research Funding (US FDA). The project was also partially supported by Office of Women’s Health/FDA (Xie and Ju), and an appointment to the Research Participation Program at the U.S. Food and Drug Administration administered by the Oak Ridge Institute for Science and Education through an interagency agreement between the U.S. Department of Energy and the U.S. Food and Drug Administration.”

Author’s Response: That you for your corrections. We have revised the Acknowledgment section in the manuscript by replacing the statement described above with the below narrative.

“Acknowledgments: We would like to thank Elliot Rosen and Dr. Su-ryun Kim for critically reviewing the manuscript.”

Please keep the funding-related text below as our Funding Statement. We have included this statement in our Cover Letter.

“This project was funded by the CDER Advanced & Domestic Manufacturing Initiatives (US FDA), the Office of Pharmaceutical Quality Centers of Excellence (US FDA), and CDER Intramural Research Funding (US FDA). The project was also partially supported by Office of Women’s Health/FDA (Xie and Ju), and an appointment to the Research Participation Program at the U.S. Food and Drug Administration administered by the Oak Ridge Institute for Science and Education through an interagency agreement between the U.S. Department of Energy and the U.S. Food and Drug Administration.”

3) Thank you for stating the following in your Competing Interests section: “None.”

Author’s Response: The authors have declared that no competing interests exist. We have included this statement in our Cover Letter. Thanks for your help to change the online submission form on our behalf. 

4) PLOS ONE now requires that authors provide the original uncropped and unadjusted images underlying all blot or gel results reported in a submission’s figures or Supporting Information files. This policy and the journal’s other requirements for blot/gel reporting and figure preparation are described in detail at https://journals.plos.org/plosone/s/figures#loc-blot-and-gel-reporting-requirements and https://journals.plos.org/plosone/s/figures#loc-prguieparing-figures-from-image-files. When you submit your revised manuscript, please ensure that your figures adhere fully to these guidelines and provide the original underlying images for all blot or gel data reported in your submission. See the following link for instructions on providing the original image data: https://journals.plos.org/plosone/s/figures#loc-original-images-for-blots-and-gels.

Author’s Response: We have provided the original images used for the immunoblots and Coomassie-stained gels in the S1 Raw images files as described in PLOS ONE’s instructions in the Blot and Gel Reporting Requirements section of the document in the following link: Figures | PLOS ONE. We also confirm adherence to the guidelines in the section “Preparing Figures from Image Files.”

5) In your Data Availability statement, you have not specified where the minimal data set underlying the results described in your manuscript can be found. PLOS defines a study's minimal data set as the underlying data used to reach the conclusions drawn in the manuscript and any additional data required to replicate the reported study findings in their entirety. All PLOS journals require that the minimal data set be made fully available. For more information about our data policy, please see http://journals.plos.org/plosone/s/data-availability.

Author’s Response: To ensure adherence to the PLOS ONE and FDA policies that all the scientific data be made publicly available, we have included an additional supporting file (S1 File) that contains the full immunoblots, Coomassie-stained gels, and the calculated mean values with standard deviations from each quantitative analytical procedure that is not currently presented in the main manuscript, including S1 Fig, S1 Table, and S2 Table documents. We have revised the Data Availability section of the main manuscript with the below statement to inform the readers that the data is publicly available.

“In accordance with the FDA and PLOS ONE policies, the data described in this manuscript is publicly available as presented in either the main figures of the manuscript or in the supporting information.”

6) PLOS requires an ORCID iD for the corresponding author in Editorial Manager on papers submitted after December 6th, 2016. Please ensure that you have an ORCID iD and that it is validated in Editorial Manager. To do this, go to ‘Update my Information’ (in the upper left-hand corner of the main menu), and click on the Fetch/Validate link next to the ORCID field. This will take you to the ORCID site and allow you to create a new iD or authenticate a pre-existing iD in Editorial Manager. Please see the following video for instructions on linking an ORCID iD to your Editorial Manager account: https://www.youtube.com/watch?v=_xcclfuvtxQ

Author’s Response: I, Tongzhong Ju, as the corresponding author have done the “Update my Information” in his ORCID iD accordingly.

7) Please include captions for your Supporting Information files at the end of your manuscript, and update any in-text citations to match accordingly. Please see our Supporting Information guidelines for more information: http://journals.plos.org/plosone/s/supporting-information.

Author’s Response: Captions to the supporting information have been added to the revised manuscript based on the style and information provided in the PLOS ONE style templates. The below level 1 title and body narrative have been placed after the Reference section.

“Supporting information

S1 Fig: Batch analyses of upstream manufacturing in ACE2-Fc and ACE2(NN)-Fc lines with fed and unfed conditions. Three separate batches with unfed and fed conditions were analyzed on days 3, 5, and 7 for (a) viable cell density (b) percentage viability (c) protein production. 

S1 Table: One- and Two-Way ANOVA results describing the degrees of freedom and p values.

S2 Table: N-glycans identified on ACE2-Fc and ACE2(NN)-Fc.

S1 File: Immunoblots, gels, and quantitative results as supportive data”

Author’s Response: We have reviewed the reference list and reformatted the list based on PLOS ONE’s current citation format.

 

The below Author’s Response address the PLOS ONE reviewer’s comments.

Reviewer #1: The article is well written and the results presented appear valid. There is a point I find to be assessed under discussion section. The activity assay for ACE2 is limited to the specific enzymatic activity. However, ACE2 is involved in a number of protein-protein interactions into the cell, whose function is at least in part unclear. The activity assay can not evaluate the ability of the recombinant fusion protein to mimic ACE2 for these interactions and activities in the cell, so this point should be considered as a risk of interference with cellular processes, with unknown effects.

Author’s Response: We appreciate the Reviewer’s comment related to the multifaceted nature of ACE2 intracellular signal transduction in cells and tissues. The rationale for using the ACE2 activity assay was only to confirm the structural integrity of the ACE2 domain within the manufactured recombinant ACE2-Fc fusion protein and comparatively demonstrate that the mutated ACE2 domain within the recombinant ACE2(NN)-Fc fusion protein caused the loss catalytic capabilities. This manuscript’s primarily focus was on the establishment of a production platform that generates ACE2 fusion proteins that can be leveraged in a follow up manuscript to address potential adverse events caused by ACE2-Fc fusion proteins using in vivo models, such as cells and animals. Therefore, at this time, mimicking ACE2 interactions and activities in cells using the ACE2 fusion proteins was out of scope for this manuscript. In a follow up manuscript, we plan to address the impact of ACE2 signal transduction in cells treated with the ACE2 fusion proteins in the presence and absence of SARS-COV to investigate this area of research that has limited scientific literature.

Reviewer #2: The authors present a well written, concise manuscript describing the use of ACE2 fused to Fc regions as an initial step in developing therapeutics that would act as a "sink" for SARS-2 and potentially SARS, though the authors make no claims about SARS. The authors demonstrate that a couple of key mutants knock out the enzymatic function which improves the usability of the constructs as virus sinks. There are a few minor typos that can be easily corrected with another round of proofreading.

Author’s Response: We would like to thank the Reviewer for his/her kind remarks and identifying typos in our manuscript. We have proof-read the original manuscript and have corrected typos, revised grammar issues, and revised sentences to further improve the manuscript’s narrative for readers in our revised manuscript

Specific comments:

Does the glycan profile vary based on cell type? If the constructs were made in human/nonhuman primate cell lines, would the binding affinities be the same?

Author’s Response: The glycosylation profile of therapeutic proteins can change based on several factors that include the host cell line used to manufacture the protein-of-interest and the overall manufacturing strategy (upstream and downstream processes). For example, a non-genetically engineered E. coli-based host cell line will produce non-glycosylated therapeutic proteins, while a Chinese hamster ovary (CHO)-based host cell line will produce “human-like” glycans on therapeutic proteins and a HEK293-based host cell line will produce proteins with human glycans. Furthermore, the glycan profiles of glycoproteins such as ACE2 also vary based on the cell types within the same species. We primarily focus on manufacturing recombinant proteins in CHO cell lines, because >70% of biotechnology products are manufactured using this host cell line. However, we are actively investigating the impact of N-glycans on the ACE2 fusion proteins by manufacturing N-glycovariants of ACE2-Fc and ACE2(NN)-Fc using a N-glyco-engineered CHO cell lines to be relevant to most therapeutic protein manufacturing practices. The Reviewer raised an excellent question. At this time, this manuscript only had to provide the foundational production platform to support future studies that will address this question, which is currently out of scope for this proof-of-concept study.

How does the ACE2-Fc-virus complex get metabolized to clear virus from the body to prevent infection?

Author’s Response: At this time, we are unaware of any scientific literature that clearly identifies the mechanisms involved in the clearance of the ACE2-Fc-viral complex. We speculate that the complex is systemically removed by mechanisms that regulate the pharmacokinetic properties of therapeutic proteins, which include receptor and non-receptor mediate lysosomal degradation. This is an excellent question, but it was out of scope for this study that built the platform for ACE2-Fc fusion protein production.

What is the significance of the alpha, beta, and delta having higher affinity than the parental or omicron.

Author’s Response: At this time, potency has only been measured using biolayer interferometry, which is a non-cell based assay that can monitor biological activities such as protein-to-protein interactions. Additional studies using a cell-based assay and an animal study are required to fully determine the significance of these differences. With this initial data, we speculate that the ACE2-Fc proteins can bind to different S protein variants, but the increase in binding affinity has the potential to impact efficacy and the effective dose of ACE2-Fc is potentially dependent on the viral variant that caused the infection.

---

## [Editor Report · Decision Letter 1]

15 Nov 2022

SARS-CoV-2 spike protein variant binding affinity to an angiotensin-converting enzyme 2 fusion glycoproteins

PONE-D-22-26644R1

Dear Dr. Ju,

We’re pleased to inform you that your manuscript has been judged scientifically suitable for publication and will be formally accepted for publication once it meets all outstanding technical requirements.

Kind regards,

Nagarajan Raju

Academic Editor

PLOS ONE
---

## [Editor Report · Acceptance letter]

21 Nov 2022

PONE-D-22-26644R1 

SARS-CoV-2 spike protein variant binding affinity to an angiotensin-converting enzyme 2 fusion glycoproteins 

Dear Dr. Ju:

I'm pleased to inform you that your manuscript has been deemed suitable for publication in PLOS ONE. Congratulations! Your manuscript is now with our production department. 

Kind regards, 

on behalf of

Dr. Nagarajan Raju 

Academic Editor

PLOS ONE